

# Spatial variability and potential maximum intensity of winter storms over Europe

Michael A. Walz[1] and Gregor C. Leckebusch[1]

[1]University of Birmingham, School of Geography, Earth and Environmental Sciences, B15 2TT, UK

**Correspondence:** Michael A. Walz (maw526@bham.ac.uk)

**Abstract.** Extra–tropical wind storms pose one of the most dangerous and loss intensive natural hazards for Europe. However, due to only 50 years of high quality observational data, it is difficult to assess the statistical uncertainty of these sparse events just based on observations. Over the last decade seasonal ensemble forecasts have become indispensable in quantifying the uncertainty of weather prediction on seasonal time scales. In this study seasonal forecasts are used in a climatological context:

By making use of the up to 51 ensemble members a broad, physically consistent statistical base can be created. This large sample can thus be used to assess the uncertainty of extreme wind storm features such as intensity or severity more accurately. In particular return periods and even a potential maximum intensity of windstorms and extra–tropical cyclones (ETCs) can be calculated depending on a specific cluster or region in Europe. A 100–year event minimum core pressure in Central Europe, for example, is estimated to be around 940 hPa, whereas it would be around 928 hPa for the British Isles. By employing extreme

value statistics a potential minimum core pressure (maximum curvature) can be estimated as well. This is way below (above) a 1000–year event however, it can therefore be seen more as a physical barrier than a realistic scenario.

## 1 Introduction and Motivation

European winter windstorms are responsible for extreme surface winds and heavy precipitation events that result in major flooding in many parts of the European domain. As a result windstorms have a vast impact on socio-economic structures of

the resident societies. Windstorm Xaver that hit Europe in December 2013 was responsible for economic losses somewhere between 700 Million and 1.4 Billion Euros (AIR Worldwide, 2013) and around 10 casualties. The extra-tropical cyclones (ETCs) instigating these windstorms have their origin over the Northwest Atlantic. They usually follow an eastward trajectory until they eventually affect the European continent. ETCs play a crucial role in the reduction of the meridional temperature gradient as they convert potential energy into turbulent kinetic energy (Leckebusch et al., 2008b). This implies that ETCs act

as a central nexus between the large scale dynamics of the atmosphere and direct local impacts, manifested in economic losses caused by associated extreme surface winds (Gaffney et al., 2007). Recent studies have shown that ETCs are steered by a variety of large scale drivers which to some degree can be used as a prognostic tool to estimate the amount of windstorms per winter season (Walz et al., 2018a, Vitolo et al., 2009, Mailier et al., 2006). This in turn means that a better understanding of the large scale variability and the properties of ETCs in general can have an enormous societal impact. A general technique

to categorise meteorological data and ultimately link it to large scale dynamics is represented by a clustering approach which





has frequently been used in the literature (e.g. Philipp et al., 2007 or Leckebusch et al., 2008b). Thus, data is assigned to different clusters so that each respective cluster contains alike data. In terms of ETCs there have been a handful of studies that have applied a clustering approach.Leckebusch et al. (2008b) used a k-means approach to classify meteorological circulation regimes which are accountable for windstorms in Europe. They could identify 4 principle circulation patterns (Primary Storm

Clusters; PSC) that are responsible for the occurrence of harmful ETCs during the extended winter season (Oct-Mar). Jointly these 4 identified clusters contain more than 70% of the historic storms between the years 1958 until 1998. Blender et al. (1997) also used a k-means clustering approach; however instead of circulation patterns they classified the trajectories of identified ETCs. The major drawback of the k-means approach in the context of clustering is its requirement for the data to be of the same length. As windstorms/ETCs occur in different spells, this approach is only partly useful for the sake of the

problem. Gaffney et al. (2007) proposed a probabilistic clustering approach in which ETCs are considered in a Lagrangian view as each of them features a unique track and life cycle. Their approach allows for trajectories to be assigned to clusters regardless of their duration. As this approach will be implemented for this study, a more detailed explanation follows in the Methods section of this paper. Owing to new Reanalysis products like the European Centre of Medium-range Weather Forecasts (ECMWF) ERA-20C (Poli et al., 2016) or the NOAA 20CR (Compo et al., 2011) there is little more than 100 years of high

quality windstorm "observations" on grid cell level. In order to estimate the uncertainty of high impact windstorms in terms of frequency and severity however, the amount of data is still too sparse to produce reasonable confidence intervals. Similar to Osinski et al. (2016) in which the ECMWF Ensemble Prediction System (EPS) is used as a data archive for creating a windstorm catalogue, this study approaches the retrospective predictions of ECMWF Seasonal Forecast System 4 (Molteni et al., 2011) as an archive of potential windstorms. Clearly none of the windstorms found in these forecasts ever happened,

however each of them represents one possible physical consistent realisation of a potential reality. Due to the 51 ensemble members of System 4 this leads to a substantial increase (around 1500 years) in the available sample of potential extreme events. This will allow for a more accurate estimation of uncertainties regarding features of extreme windstorms, e.g. intensity or duration. The ensemble members are treated as statistically independent since studies have shown that there is little forecast skill for very high local wind speeds (>98th percentile; Walz et al., 2018b). As predictability indicates statistical dependence

of the ensemble members, the inverse is also true; hence no/little predictability implies statistical independence of the original ensemble (DelSole and Tippett, 2007, DelSole, 2004). The novel approach of this study is to gain a better understanding of the hazard uncertainty of windstorms and ETCs by utilizing the ensembles of seasonal retrospective forecast data in a climate archive approach. Events will be identified based on two different objective tracking algorithms, one of which is based on wind speed whereas (Leckebusch et al., 2008a or Kruschke, 2015) the other one is based on maxima in curvature of the mean

sea level pressure (MSLP) field (Murray and Simmonds, 1991). The events will be classified based solely on the shape of their respective trajectory. The different clusters will be analysed with regard to storm features such as maximum intensity, duration and celerity. Eventually a most and least intense storm cluster will be identified. Section 2 will describe the ECMWF System 4 data which is used for this study. Section 3 summarizes the method of the probabilistic clustering approach proposed by Gaffney et al. (2007), before Section 4 presents the results of this approach. The paper will close with a Summary and

Discussion in Section 5.



## 2   Data

The idea of this study is to utilize the ECMWF System 4 (Molteni et al., 2011) as a climate archive in a way of assuming the 51 members of retrospective forecasts each resemble an artificial reality. System 4 was the operational seasonal forecast system until November 2017. This study uses retrospective forecasts which are initialised at November $1^{st}$ each year from 1983 until

2013. Every run lasts for 7 months so that data from November $1^{st}$ until May $31^{st}$ are available. Due to the spin-up (avoiding potential "real storms" at the beginning to guarantee statistical independence) of the model and the focus on European winter windstorms only, the months December until March are used for this study. The ensemble entails 51 members which, combined with 31 years of data, is equivalent to 1581 "virtual" years or winter seasons respectively. That way the ensemble serves as a unique data archive which can be used to assess the statistical uncertainty more precisely compared to exploiting observational

reanalysis data for this kind of estimation. System 4 is provided on a spectral resolution of T255 which is the same resolution used for the Reanalysis ERA Interim. The perturbed initial conditions are produced using singular vectors and an ensemble of ocean conditions of the ocean model NEMO (Madec et al., 2015). The forecast system is based on the IFS cycle 36r4 of the ECMWF. Two different types of identification methods are used for this study: Windstorms are identified using an objective windstorm tracking algorithm that is based on the exceedance of the local 98th percentile of wind speeds (Leckebusch et al.,

2008a; Kruschke, 2015). Cyclones are identified using the cyclone identification method developed by Murray and Simmonds (1991) which is based on finding maxima in the curvature (c) of the MSLP field. Both times the tracking is carried out for the entire Northern Hemisphere, however only events affecting specific countries/areas of Europe are analysed within this study. Due to the abundance of windstorms and cyclones it is possible to determine the hazard uncertainty of windstorms on a country/region level. This is implemented by only taking into account windstorms that affect a country at least once in their

lifetime, i.e. by defining a radius around a country/an area through which a windstorm or a cyclone has to pass. Thus tracks, that never crossed a respective area (e.g. British Isles) are discarded from the analysis The area of the maximum wind field of a cyclone is usually found southeast of the core of the cyclone (Fig.1; Leckebusch et al., 2008a). That is why the selection radii for the tracked cyclones are slightly different to the windstorm ones, i.e. shifted a bit towards the northwest. As the cyclones identified by the Murray and Simmonds algorithm are not necessarily extreme in terms of impact, the minimal core pressure

and the maximum curvature of an identified track both have to be within the lowest respectively highest 5% of all tracks at least once within the defined radius. This constraint reduces the number of cyclone tracks significantly. However, it represents a necessary approach since a cyclone that is very intense somewhere over the Atlantic is unlikely to embody a severe damage potential for Europe. For illustration purposes of the tracking algorithm Figure 1 depicts a snapshot of the wind field as well as an identified cyclone track (red) together with its matched windstorm track (black). The matching of the two events was

carried out according to the algorithm implemented by Nissen et al. (2010). The MSLP field is depicted as a black overlay. As expected the maximum wind speed can be found just south of the area of minimum core pressure of the cyclone. Evidently the cyclone trajectory is significantly longer than the windstorm track. This is due to the fact that a windstorm per definition is only tracked as long as the local 98th percentile of wind speeds is exceeded whereas the cyclone is tracked over its entire lifetime from cyclogenesis to cyclolysis. In total more than 11,000 windstorms could be identified that fulfilled the area criterion for





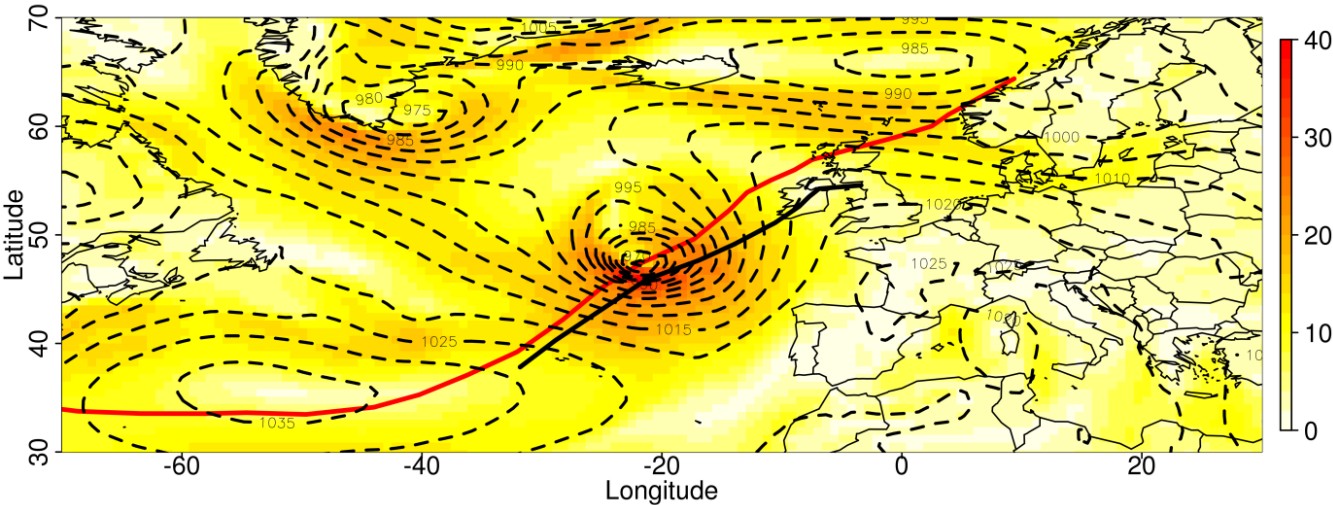

**Figure 1.** Instantaneous wind speed of an arbitrary identified windstorm in System 4, overlayed by the MSLP field in black dashed lines. The red line represents the entire identified cyclone trajectory, the black line the entire associated windstorm track. As expected the area of maximum wind speeds is south of the core of the cyclone.

the British Isles amongst the 1581 virtual years of data which is an average of around 7 windstorms per December-March period. This compares to about 7,000 storms affecting Germany and the Benelux, around 8,000 for the Scandinavian region and approximately 15,000 affecting the Central European region. The exact numbers can be found in Table 1 in Section 4.1. Due to the constraint for the tracked cyclones the number of identified events is distinctively smaller. There are around 3,000 extreme cyclones affecting the British Isles, around 700 for Germany and the Benelux, 2300 for Scandinavia and roughly 1000 for Central Europe. Even though the numbers are smaller compared to the number of windstorms for every region, the sample size is still large enough to apply extreme value statistics. Clearly there is also an overlap between the different regions; however there are always windstorms/cyclones that are unique for each region. Additionally the overlap does not affect the estimation of the uncertainty within each particular region.

## 3 Methods

### 3.1 Clustering technique

In order to categorize the identified trajectories for cyclones and windstorms the regression mixture models clustering method proposed by Gaffney et al. (2007) was implemented in FORTRAN. This paper will only provide a short summary of the method, for a full description of the subject the reader is referred to their original study. The initial position of every identified trajectory is subtracted from each respective pair of longitude and latitude coordinates in order not to cluster the tracks based on their origin but solely based on their shape. The initial location of a cyclone might influence the shape of the track, as





we discard tracks, however, that never make landfall in Europe we can assume all of the tracks are shaped "regularly". The amount of resulting clusters has to be defined prior to the clustering. In agreement with Gaffney et al. (2007) three clusters are chosen to classify the windstorm/cyclone tracks. This choice is based on their results as well as on some qualitative inspection. Additionally three clusters provide a coarse overview as this study tries to understand the big picture of cyclone tracks, whereas

with many clusters the clustering becomes more and more fuzzy. The trajectories are modelled via a second order polynomial which was determined to perform best via cross-validation and which is also in accordance to Gaffney et al. (2007). The concept of the probabilistic clustering with regression mixture models is firstly to learn all the parameters (regression and covariance matrices) for all $K = 3$ clusters and in a second step to decide in which cluster a respective trajectory is most likely to be in. In other words, every trajectory is fitted by a quadratic polynomial and based on the coefficient matrices of

this polynomial the algorithm calculates probability weights $w_{ik}$ with $\sum_{k=1}^{3} w_{ik} = 1$. Subsequently these weights decide in which of the $K = 3$ clusters a respective track is most likely to be in. Starting from a random initial probability for every trajectory, three regression matrices $\beta_k$ of size 3x2, 3 covariance matrices $\sigma_k$ of size 2x2 and a probability $w_{ik}$ is computed by the Expectation-Maximation (EM) algorithm. The EM algorithm is widely used for estimating the maximum-likelihood parameters in connection with regression mixture models (Dempster et al., 1977, McLachlan and Krishnan, 2007). It is a two-

step algorithm as initially both of the parameters ($\beta_k$ and $\sigma_k$) as well as the cluster assignment are unknown. The algorithm is iterated until the increase in the maximum-likelihood estimation falls below a certain threshold. A drawback of the EM algorithm is its potential to only find a local maximum in the maximum-likelihood surface. To increase the chance of finding a global maximum the algorithm is run 50 times with 50 different random starting weights. The events are hard-clustered by assigning a trajectory to the cluster featuring the largest of the three probability weights ("winning weight").

## 3.2    Analytical techniques – Windstorms

After assigning every windstorm trajectory to one of the three clusters the dynamical features of the associated events are examined. The intensity of a windstorm is given by the Storm Severity Index (SSI), an objective measure for the severity of a storm based on the cubic exceedance of the local 98th percentile of wind speeds (Leckebusch et al., 2008a). The SSI is part of the output of the windstorm identification and tracking algorithm. It is calculated and accumulated on a grid cell level

for every time step of a respective windstorm. In order to assess the damage potential of a storm for a specific region, only SSI values within the defined radius around a region are added up. By applying means of extreme value statistics, i.e. fitting of a Generalised Pareto Distribution (GPD, e.g. Coles, 2001) to the excesses over a large threshold of SSI values (Peak over Threshold (POT) approach), return levels of windstorm intensities for every region and cluster can be estimated. The POT approach has been adapted by many other studies in connection with excessive precipitation (e.g., Vrac and Naveau, 2007 or

Cooley et al., 2007), wind speeds (Kunz et al., 2010) and also SSI values (Donat et al., 2011 or Held et al., 2013). The confidence intervals of the return levels are estimated via profile likelihood (Coles, 2001) as the intervals become highly asymmetric for upper bounds and high return periods. The parameter estimation of the GPD and the calculation of the confidence intervals are implemented in R with the help of the *ismev* library (Heffernan et al., 2012). In order to account for the probabilistic nature of the clustering, the cluster weights are used to weight features of the events associated with each trajectory. If, for example, the



winning weight was 0.75 an SSI value of 1 will result in a weighted severity of 0.75. Given the shape parameter of the GPD distribution is negative; an expected maximum value of SSI values can be estimated. Additionally some general storm features like celerity, duration and average intensity are estimated for every cluster and region. MSLP anomaly composites are created as well in order to gain insight in the large scale conditions that predominate for the three clusters. This is done by averaging
the anomaly MSLP fields of the windstorm days for the three clusters.

### 3.3  Analytical techniques – ETCs

The intensity measures of cyclones are denoted by the curvature of the MSLP field $c$ and the minimal core pressure $p$ of the cyclone. Both features do not guarantee a potential high-impact storm event, however they both serve as very good proxies. As only cyclones are selected that range among the top 5% of all cyclones in terms of p and c within the defined areas around the
regions, the chances are increased for the cyclone to have a severe impact in terms of wind speeds. Similar to the procedure applied to the windstorms events, a GPD is fit to the distribution of excesses of each c and p. That way return levels of minimum core pressure and maximum curvature as well as their uncertainties can be estimated for every region and cluster. These maximum/minimum values can be compared and a cluster of the highest intensity potential can be determined for every region. Furthermore, a potential upper/lower limit of the curvature/core pressure and their uncertainty can be estimated. For a
general overview of the identified clusters common attributes of cyclones like celerity, average core pressure and duration are examined and compared amongst the clusters and for the different regions.

## 4  Results

### 4.1  Windstorms

The three identified clusters for the windstorm trajectories can be seen in Figure 2 and 3. Figure 2 illustrates windstorms
affecting the British Isles (BI), whereas Figure 3 depicts storms for Germany and the Benelux (GEBE) countries. The clusters for the other two regions look very similar (not shown). Clearly there are three general directions of progression of the tracks. The first cluster features events that follow the well-known North Atlantic storm track crossing the Atlantic in a north-easterly direction. For all the windstorm tracks it has to be kept in mind that the actual cyclone track might be a lot longer as windstorms by definition only exist whilst the 98th percentile is exceeded. Cluster 2 includes events that are generally shorter than the ones
in the first cluster: Their paths reflect a straighter West-East progression across the Atlantic compared to the first cluster. Events in the second cluster tend to be detected later, thus more towards the Central Atlantic. From there they proceed in an almost straight track towards the East. This cluster also contains some events that are only detected when the system is already close or even within the defined radii, i.e. the British Isles. Events that are included in Cluster 3 look distinctively different compared to the other two identified clusters. Events featured in cluster three are identified over the Denmark Straight usually between
Greenland and Iceland or just south of Iceland. The tracks then follow a south-easterly trajectory, approaching Europe by crossing the Northern Sea. This is also the cluster that differs most from the clusters identified by Gaffney et al. (2007). Their





clusters D and H resemble Cluster 1 and 2 of this study whereas their cluster V cannot be identified for the tracks of System 4 at all. The progression of the tracks within these clusters is almost reversed: south-north compared to northwest-southeast.

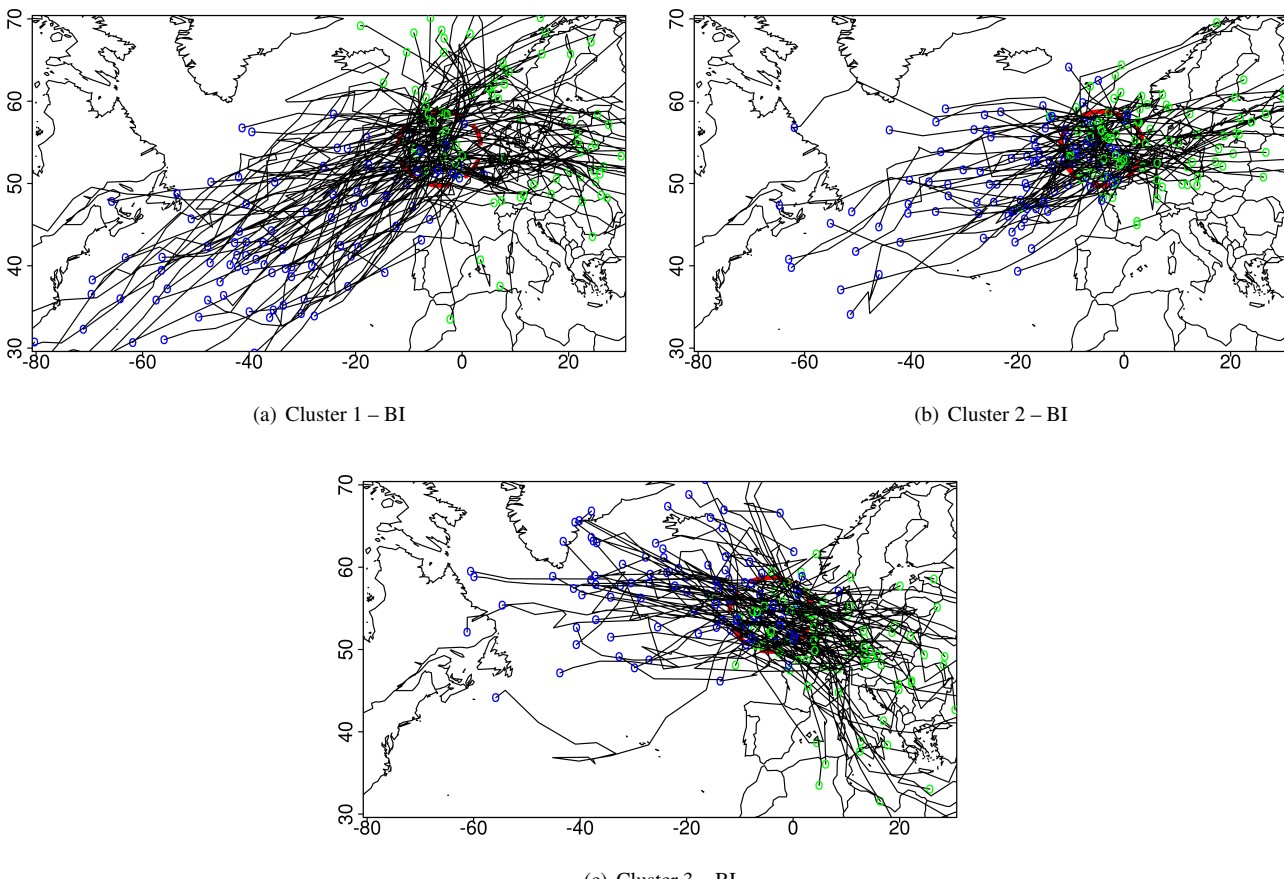

(a) Cluster 1 – BI

(b) Cluster 2 – BI

(c) Cluster 3 – BI

**Figure 2.** The three clusters that were identified by the probabilistic clustering method for windstorms affecting the British Isles. Cluster 1 (top left) with its southwest to northeast progression, Cluster 2 (top right, west to east) and Cluster 3 (bottom, northwest to southeast). 100 random tracks are shown for each of the clusters. The blue points mark the beginning of the track whereas the green ones depict the last time step of each trajectory. The red circle encloses the British Isles and has to be crossed at least once in the life cycle of the windstorm.

This is certainly due to the fact that for the present study only events were considered that actually affected one of the defined regions, whereas Gaffney et al. (2007) clustered all tracks that could be identified for their entire domain (i.e. have their origin further west). The number of storms for every region and cluster can be found in Table 1. Clearly the Cluster 2 is the most frequent across all 4 regions as it makes up around 50% of all storms except for the Scandinavian region where it accounts only for about 40% of all the storms. Cluster 1 and 2 feature similar numbers for 4 all regions with Cluster 1 being more frequent in the BI and GEBE whereas Cluster 3 is more frequent in Central Europe (CE) and Scandinavia (SC).





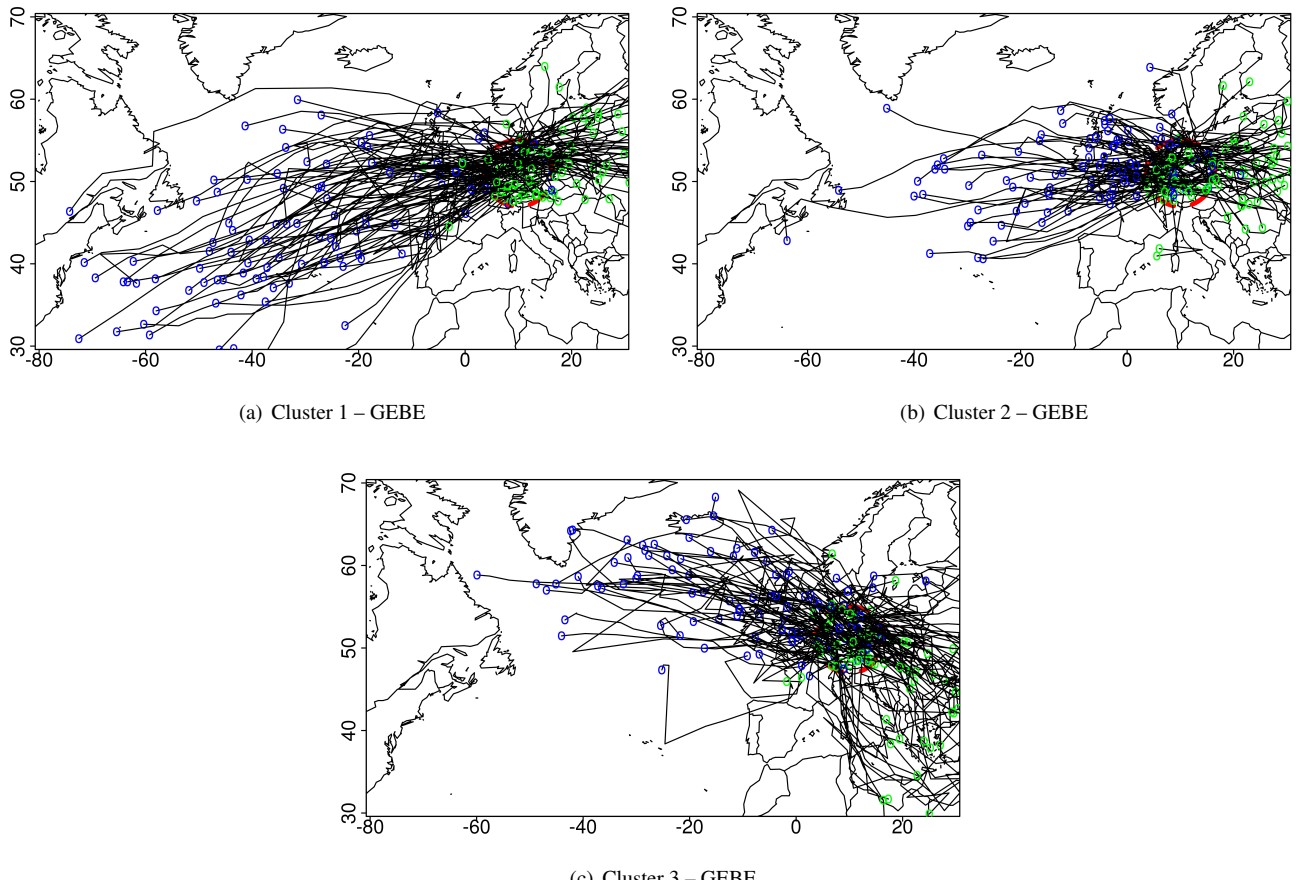

(a) Cluster 1 – GEBE

(b) Cluster 2 – GEBE

(c) Cluster 3 – GEBE

**Figure 3.** Same as above except for windstorms affecting Germany and the Benelux (GEBE).

**Table 1.** Number of windstorm tracks for all of the three clusters for the four different regions studied. The arrow behind the clusters represents the general path of the trajectory for each cluster.

| Region | Cluster 1 ↗ | Cluster 2 ⟶ | Cluster 3 ↘ | Total |
|---|---|---|---|---|
| British Isles | 3182 | 5374 | 2482 | 11038 |
| Germany and Benelux | 1599 | 3911 | 1512 | 7022 |
| Central Europe | 3471 | 7244 | 4182 | 14897 |
| Scandinavia | 2318 | 3220 | 2863 | 8401 |





Figure 4 provides (weighted) SSI return level estimates and their uncertainties for every region and cluster. Overall Cluster 1 represents the cluster with the most intense storms especially for the lower return periods. Particularly for the BI and GEBE the SSI return levels for Cluster one are the largest values for all periods, making it the most hazardous of the 3 clusters in terms of potential damage for these two regions. In the same way Cluster 3 appears as the least hazardous cluster as it contains

the lowest SSI return levels across all return periods. The difference in magnitude between the clusters can be quite substantial as for example a 100 year event for the British Isles within Cluster 1 would be a 200 year event for Cluster 2 or almost a 500 year event for Cluster 3. Clearly the uncertainty of Cluster 3 is the lowest amongst the three clusters for BI and GEBE. Even intervals for the very large return periods appear almost symmetric around the estimated value whereas the confidence intervals for the high return levels of Cluster 1 and 2 are highly asymmetric towards the larger values. This reflects the larger uncertainty

towards very high impact windstorm events. For CE and SC the order of the most hazardous cluster appears slightly different. For lower return periods Cluster 1 still represents the most hazardous cluster, however for the 500-year and 1000-year return level Cluster 2 for CE and Cluster 3 for SC emerges as the most hazardous one, especially when considering the confidence intervals. For the 50-, 100- and 200-year return period, Cluster 2 and 3 are virtually identical for CE. Interestingly Cluster 3 for SC is potentially more hazardous for return levels including the 500-year period than Cluster 2 which, in contrast to all other

regions, represents the least hazardous cluster for SC. Overall, return levels for the 1000-year period for CE and SC look very similar across all three clusters. Opposed to the BI and GEBE, Cluster 1 represents the cluster with the least uncertainty whereas Cluster 2 appears as the most uncertain, particularly for Central Europe where the upper confidence interval for the 1000-year return level reaches up to a value of 63. This is due to extremely intense outliers within Cluster 2 and also Cluster 3. Owing to the amount of storms per cluster and region however, the confidence intervals are still comparatively small considering the very

large nature of the return periods. Generally SSI values over the mainland of Europe are systematically larger than over the BI and parts of Scandinavia. This is a result of the construction of the SSI: As the definition of the SSI does not take into account the shape of the distribution of wind speeds past the 98th percentile, SSI values are systematically higher for grid cells in which windstorms occurrences are less frequent (Walz et al., 2017). From an impact perspective this assumption is valid as it can be expected that the infrastructure in these areas might not be as adapted to frequent windstorm events as it is in areas within the

main storm corridor (i.e. South of France vs. the British Isles). Basic (weighted) storm features are compared by kernel density plots which are presented in Figures 4 (BI) and 5 (GEBE). The analogous figures for CE and SC are omitted at this point as the basic features for all three clusters look very similar. Clearly Cluster 1 is the most extreme cluster in all three features addressed. Particularly the celerity of the events in Cluster 1 is considerably larger than for the other two clusters which appear very similar. Cluster 2 features by far the shortest events. This confirms the impression that could already be drawn from the

trajectories in Figure 2. The larger area of windstorms in Cluster 1 partly explains the increased severity of windstorms as the numbers examined in Tables 2 and 3 are integrated values over all grid cells affected within the radius around a particular region. Interestingly Cluster 1 also entails the largest events in terms of area for CE and SC. As the largest return level for the 1000-year event is found for Cluster 2 however, this means that there is no linear relationship between area and intensity of a windstorm event. Figure 5 provides the weighted windstorm features for GEBE. The overall attributes are very similar

compared to the BI across all clusters with windstorms in Cluster 1 being the largest and fastest traveling events. Events from





Cluster 2 and 3 however exhibit an increased average celerity compared to the same clusters for the BI. The meteorological conditions that are predominant during windstorm events for each cluster are presented in Figure 6. The panel shows MSLP anomaly composites of all the days on which windstorms were identified for each cluster for the BI region. All three figures show the stationary low pressure system that is associated with the Icelandic Low. The shape and location of this system, however, is different for all three clusters. Whereas for Cluster 1 and 2 the centre of the MSLP composite lies southwest of Iceland, the centre for Cluster 3 is shifted towards the northeast. This explains the different trajectories for the major part of storms in Cluster 3 compared to the other two clusters as the MSLP gradient points in a different direction. The direction of the MSLP gradient for each cluster is approximately equivalent to the main path of the trajectories within each cluster, thus the direction of the large scale atmospheric flow under the geostrophic wind approximation. Windstorms in Cluster 3 appear upstream ("backside of the wave") of the atmospheric wave, whereas windstorms in the other two clusters are the result of a more downstream development. Compared to the quiescent composites (not shown) the gradient is considerably stronger explaining the higher potential for storminess for each of the three identified clusters. The MSLP patterns for the windstorms affecting the CE region are presented in Figure 7. The overall MSLP anomalies look similar to what was shown for the BI region, however the minimum of the anomalies is shifted further to the East, for Cluster 1 in particular. In accordance to the findings for the cyclones (see Chapter 4.2), events in Cluster 1 are lower in core pressure compared to the other two clusters (-18 hPa vs. -14 hPa; compare the following chapter). The MSLP composites for the other 3 regions are according to the one shown for the BI and CE. The associated MSLP anomalies for the windstorms affecting the CE region draw a similar picture, even though the Icelandic Low is shifted more towards the South. In view of the more southern region of CE this is according to the expectation. The composite for Cluster 3 emphasises the shifted low pressure system even more as the core of the system is located over southern Scandinavia which enables windstorms to travel in a south-easterly direction upstream of the associated low pressure system.

## 4.2 ETCs

The 3 identified clusters for the cyclone trajectories only differ marginally from the clusters found for the windstorms and are presented in Figure 8. However by classifying the cyclones in three different clusters we can give estimates of a *minimum* (*maximum*) core pressure (curvature) for different return periods based on a cluster. Depending on the shape parameter of the GPD fit we can even estimate an absolute to be expected minimum (maximum) of core pressure (curvature). This way a we can estimate both the most extreme cluster and the most uncertain cluster with regards to extremity.

Whereas Cluster 1 looks very similar, the trajectories in Cluster 2 and 3 look slightly different to the ones for the windstorms. As windstorms are only identified and tracked if the local 98th percentile of wind speeds is exceeded, they tend to be considerably shorter than cyclone tracks. Thus, they are usually identified at a later stage of the development of a cyclone. That is the reason why many of the cyclone tracks in Cluster 3 appear much longer than the equivalent trajectories for windstorms. Most of the windstorms in Cluster 3 have their origin in the North Atlantic and the Labrador Strait. The origin of the associated cyclone, however, is further upstream off the coast of Newfoundland. Hence, most cyclone tracks in Cluster 3 travel in a





north-easterly direction whilst they intensify until they reach the Labrador Strait. This is the point where they will be detected by the windstorm tracking algorithm. Subsequently they travel in a south-easterly direction (upstream of the wave) which leads them towards the European mainland. Similarly, cyclones in cluster 2 are identified long before the windstorm tracking algorithm associates the respective windstorm to the cyclone. In this case they are only identified as windstorms halfway across

the Atlantic Ocean which lets them appear so much shorter compared to the entire cyclone track. Generally, it is harder to distinguish between cyclone Cluster 1 and 2 both qualitatively (i.e. Figure 8) and quantitatively as demonstrated by Figure 9 and 10. Figure 9 provides return levels for the same return periods as previously for minimum core pressure of a cyclone within the defined area for BI and CE. The return levels for Cluster 1 and 2 look very similar for both regions, especially for CE where they are virtually the same throughout all return periods. In accordance with the intensities of windstorms, Cluster 3 features

the potentially least strong events, revealed by the largest minimum core pressure. Table 2 provides an estimate of the lowest potential minimum cyclone core pressure to be expected for each cluster. This estimation is based on the (negative) shape and scale parameter of the respective GPD that was fit for each cluster. As before Cluster 1 and 2 feature virtually the same value whereas Cluster 3 exhibits a larger, thus less intense, core pressure. It is not possible to calculate a lowest potential core pressure for Cluster 3 for CE as the shape parameter of this particular GPD is positive which means that no upper limit can be

estimated. Due to the abundance of cyclone events, the uncertainty in estimating the return values is fairly low considering the scarcity of these events. The confidence interval for a 500-year return level for BI and Cluster 1 for example only ranges from 919-925 hPa which is a fairly accurate estimation. Return levels of the curvature of the MSLP field c are presented in Figure 10. Generally these values are in accordance with the results for the minimum core pressure and also the intensity of windstorms. Events in Cluster 1 feature the highest curvature return levels, whereas the ones in Cluster 3 appear as the lowest. Compared to

the core pressure values however, there is a larger difference between Cluster 1 and Cluster 2. Table 2 also features estimates of the expected upper limit of c. Interestingly Cluster 3 features the largest upper limits even though the return levels were lower than both of the other clusters. One of the reasons for that could be the larger uncertainty of Cluster 3 compared to the other two clusters. Especially the uncertainty in Cluster 1 is remarkably low. Even for a 1000-year event the confidence interval comprises the estimate really closely. Additionally the confidence interval is only slightly asymmetric, whereas the confidence

**Table 2.** Potential minimum core pressure (in hPa) and maximum curvature (in hPa/deg.lat$^2$) for all three clusters for the BI and GEBE region

| Region | Cluster 1 ↗ | Cluster 2 ⟶ | Cluster 3 ↘ |
| --- | --- | --- | --- |
| British Isles | 911 \| 6.2 | 909 \| 6.4 | 918 \| 7.3 |
| Germany and Benelux | 909 \| 5.8 | 907 \| NA | NA \| 7.3 |

intervals for the 1000-year windstorm intensities are considerably asymmetric (e.g. Figure 4). In terms of intensity the events that affect the BI are clearly stronger compared to the ones affecting CE. This is in contrast to the intensities of the windstorm events as the largest potential SSI values are estimated for CE and the lowest for the BI. As mentioned before, this is in line





with the findings from Walz et al. (2017). Furthermore there is not necessarily a linear relationship between a large curvature, a small core pressure and high surface winds. Some of the windstorms might not be extreme in terms of pressure and curvature and vice versa.

## 5    Summary and Discussion

The probabilistic clustering technique proposed by Gaffney et al. (2007) was implemented for the purpose of investigating the statistical uncertainty of extreme extra-tropical cyclones. Two different identification and tracking algorithms for identifying extreme cyclone/windstorm events were applied: one based on exceedances of the local 98th percentile of wind speeds (Leckebusch et al., 2008a), the other one based on maxima in MSLP curvature (Murray and Simmonds, 1991). The events were identified in retrospective forecasts of the ECMWF Seasonal Forecast System 4 that comprises 51 ensemble members. The 51 members are used in a climate archive context similar to Osinski et al. (2016), thus every member is treated as an individual artificial reality of 31 years resulting in more than 1500 years of physically consistent data. As the forecast skill for very high local wind speeds is relatively small (Walz et al. (2018b)), the members can be seen as statistically independent since statistical dependence and forecast skill can be seen as equivalent. The clustering was implemented in accordance with Gaffney et al. (2007) who used three clusters and quadratic polynomials to model the individual trajectories. Cluster 1 and 2 are equivalent to the clusters identified in their paper. The trajectories assigned to Cluster 3, however, are entirely different as it features windstorm events that approach Europe in a south-easterly direction whereas their third cluster mainly features events of south-north displacement. On the one hand this could be due to the different tracking algorithm (wind based vs. pressure based). As mentioned above, windstorm tracks are only about half as long as cyclone tracks. This affects their shape and thereby the coefficients of the polynomial that is used to fit them. On the other hand the different cluster might be due to the geographical selection process of the windstorms used for the clustering. Gaffney et al. (2007) use identified cyclones from the entire North Atlantic area while this study only considers trajectories that cross a certain defined area, i.e. the British Isles. Thus, windstorms that do not affect any of the defined regions and that potentially travel in a south to north direction are generally excluded from the clustering algorithm. The statistical analysis was carried out via a GPD model which is part of the theory of extreme value statistics (Coles, 2001). Return periods of windstorm intensities (quantified by the SSI) for all three clusters and all four defined regions were determined. Cluster 1, which contains storms that cross the Atlantic diagonally following the classic Atlantic storm track, appears as the cluster that includes the potentially most severe windstorm events, in particular for BI and GEBE. Due to some very intense events included in Cluster 1 the uncertainty of the cluster for the very large return periods (>=500 years) is considerably larger than for the other two clusters. Considering the very large return periods, however the overall absolute uncertainty for BI and GEBE is fairly small. Especially the confidence intervals for Cluster 3, which contains the potentially least severe windstorms, are reasonably small and also symmetrical. These exact intervals can be achieved by utilizing more than 11,000 windstorms for the BI and more than 7,000 for GEBE. The intensities for CE and SC are similar; however the 1000-year event for Cluster 2 is larger than the equivalent value for Cluster 1. In the same way the confidence intervals are larger and less symmetrical compared to the ones for the BI and GEBE. Both of this is due to some outliers in



intensity being part of Cluster 2. Additionally, the differences in intensities between the clusters are generally lower, especially for SC where the 1000-year event is essentially the same across all three clusters. The intensity values are generally difficult to compare across the four regions as the SSI is dependent on the local wind speed climatology Walz et al., 2017). Large SSI return values for CE or GEBE can, however, be interpreted as a lack of preparedness of the local infrastructure in some of the

countries included in the region against these high-impact storms due to their infrequent occurrence. Despite these discrepancies between the clusters, the return levels provide a valuable quantification of the severity and especially their uncertainty for the four different regions and three different clusters. In terms of windstorm features Cluster 1 contains the largest and also longest events. As the intensity is an aggregated quantity this would explain the more severe events in this cluster. However, even though Cluster 2 contains the smallest and shortest events also for CE and SC their intensity is higher for those regions.

Thus there is not a clear linear relationship between area, duration and intensity. Regarding the celerity of the events, there is a large difference between Cluster 1 and the Clusters 2 and 3. Events in Cluster 1 travel considerably faster than the ones in the other two clusters. The MSLP anomaly composites for the three different clusters indicate a successive eastward displacement of the Icelandic Low for Cluster 1, 2 and 3 respectively which allows for the cyclones and the associated windstorms to develop more upstream, thus pursuing a south-eastward track compared to a northward movement. Compared to composites of days

without any storm activity (quiescent cluster), the MSLP gradient over the North Atlantic is substantially stronger resulting in more intense (i.e. deeper core pressure) cyclone systems that get eventually identified as a windstorm. The associated cyclones for windstorms in Cluster 1 are distinctively lower compared to the other two clusters. This is in good agreement with the findings for the tracked cyclones as well. Compared to the composites created in Gaffney et al. (2007) no negative NAO conditions are evident for Cluster 2 (their H-Cluster). This, however, is most likely due to the fact that windstorms hardly occur

during negative NAO conditions (Donat et al., 2010). The Clusters 1, 2 and 3 link to the Primary Storm Cluster (PSC) 1, 2 and 4 in Leckebusch et al. (2008b) respectively. The PSC 3 of their study can be considered as a hybrid of Cluster 1 and 2. They assigned more than 70% of the historical extreme storms to these four clusters. This in turn means that the clusters identified in this study proof to be a valid characterization of the spatial variability of extreme windstorms. The three identified clusters for the cyclone tracks are less distinct compared to the windstorm clusters. Qualitatively it is more difficult to find a difference be-

tween Cluster 1 and 2 as the trajectories of both clusters are fairly similar. However, similar to Cluster 2 for the windstorms, the trajectories in Cluster 2 for the cyclones tend to be shorter and pursue a more direct eastward track compared to the very distinct diagonal track of Cluster 1. Cluster 3 of the cyclones is comparable to Cluster 3 of the windstorms as the general progression of the identified tracks is also from the central North Atlantic towards the southeast. In contrast to the windstorm in Cluster 3 however, their origin is further west as windstorms are identified later in the life cycle of the associated cyclone. For that reason

the cyclone trajectories in Cluster 3 follow are more "arc-like" path compared to the diagonal displacement of windstorms. The intensity of the cyclones is represented by the minimum core pressure and maximum curvature. As the cyclones identified by the tracking algorithm are not necessarily extreme per se, cyclones have to be amongst the most extreme 5% in both variables at least once within the defined areas around the four regions. Even though that does not guarantee a high-impact storm it does serve as a good proxy for an extreme event. This constraint reduced the amount of cyclones for every region drastically so

that roughly around 500-1000 cyclones per region were considered. Bearing in mind the scarcity of these extreme cyclones





however, the sample size was still large enough to estimate return levels for very large return periods accurately. In accordance with the return levels of the windstorm events, the lowest core pressure can be assumed to occur in a cyclone from Cluster 1. Even though there are fewer events per cluster compared to the windstorm clustering, the uncertainty is considerably smaller (±1%) for both the minimum core pressure and the maximum curvature. Additionally the confidence intervals are fairly sym-

metrical which in turn is an indication for well-estimated confidence intervals. The reason for this good estimate is the lack of extreme outliers that made it easier to find a threshold and fit a GPD to the distribution of excesses. Economou et al. (2014) estimated core pressure extremes based on a statistical model. Their estimate for a 50–year event for the UK is approximately between 940–970 hPa (for a strongly positive NAO phase). This is higher than the estimates we could give based on System 4 (around 932-937 hPa depending on the cluster). However "only" 31 years of observational data were used for their study, so

a reason for the higher estimates (and larger uncertainties) might just be the almost factor 50 times smaller original sample size.

The lowest ever recorded MSLP in the British Isles was 926 hPa in January 1884 (MetOffice, 2016) which would, depending on the cluster, represent a 500-1000–year event based on our analysis. This shows that the estimations made from System 4 lie within the physical possible horizon. The lowest potential core pressure of a cyclone is estimated to be around 910 hPa for both

the BI and CE. This is more than 10 hPa below the 1000-year return level (and thus the lowest ever recorded value) so it can be assumed that an event of this magnitude is highly unlikely and should be regarded as a physical barrier rather than an actual event. Interestingly the largest potential curvature for both regions is found for Cluster 3 even though the return levels are the smallest of the three clusters. However, this is in accordance with the generally higher uncertainty of Cluster 3, expressed by the larger confidence intervals for both curvature and core pressure.

The clustering approach represents a useful instrument to classify these rare extreme events and to determine large-scale differences between the different clusters. Due to the approach of considering the 51 ensembles of the retrospective seasonal forecast of the ECMWF System 4 as a vast statistical base, the uncertainty in intensity and extreme core pressure even for very large return periods could be estimated fairly accurate. For most regions an empirical physical barrier of core pressure and curvature could be estimated as well. As these barriers are well below/above the 1000-year return level, they act as an estimate

of what could possibly happen rather than as a magnitude that is likely to occur. Windstorms/cyclones in Cluster 1 pose the largest threat in terms of potential damage for most of Europe as it features the largest return levels for all metrics addressed: Windstorm events in Cluster 1 are of the biggest size and of the highest celerity, thus making it the most hazardous cluster of the three.

*Data availability.*  All the data/results of this paper were created with ECMWF System 4 (Molteni at al., 2011) data which are freely available

for academic use through the ECMWF server via https://www.ecmwf.int/en/forecasts/accessing-forecasts/order-historical-datasets (MARS access)



*Competing interests.* The authors declare that they have no conflict of interest

*Acknowledgements.* M. A. Walz has been supported by a NERC CENTA DTP PhD award kindly funded by Research Council UK. The authors thank the ECMWF for providing the Seasonal System 4 data.




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



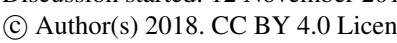

(a) Cluster 1

(b) Cluster 2

(c) Cluster 3

**Figure 4.** Weighted return levels for given return periods of SSI values for all four regions. The 95% confidence intervals are marked with whiskers and are calculated via the profile likelihood method.



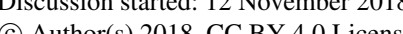

(a) Area

(b) Celerity



(c) Duration

**Figure 5.** Weighted windstorm features found for the three defined clusters for the British Isles region. Area of the tracked windstorm in units of 10,000km2 (left), average celerity of the storm in km/h (centre) and duration of the windstorm events per cluster in days (right).





(a) Cluster 1

(b) Cluster 2

(c) Cluster 3

**Figure 6.** MSLP Composites of windstorm days for each of the three clusters for storms affecting the British Isles. Cluster 1 (top left), Cluster 2 (top right) and Cluster 3 (bottom). All grey shaded areas are significant at the 99% interval





(a) Cluster 1

(b) Cluster 2

(c) Cluster 3

**Figure 7.** Same as in Figure 6 but for Central Europe.


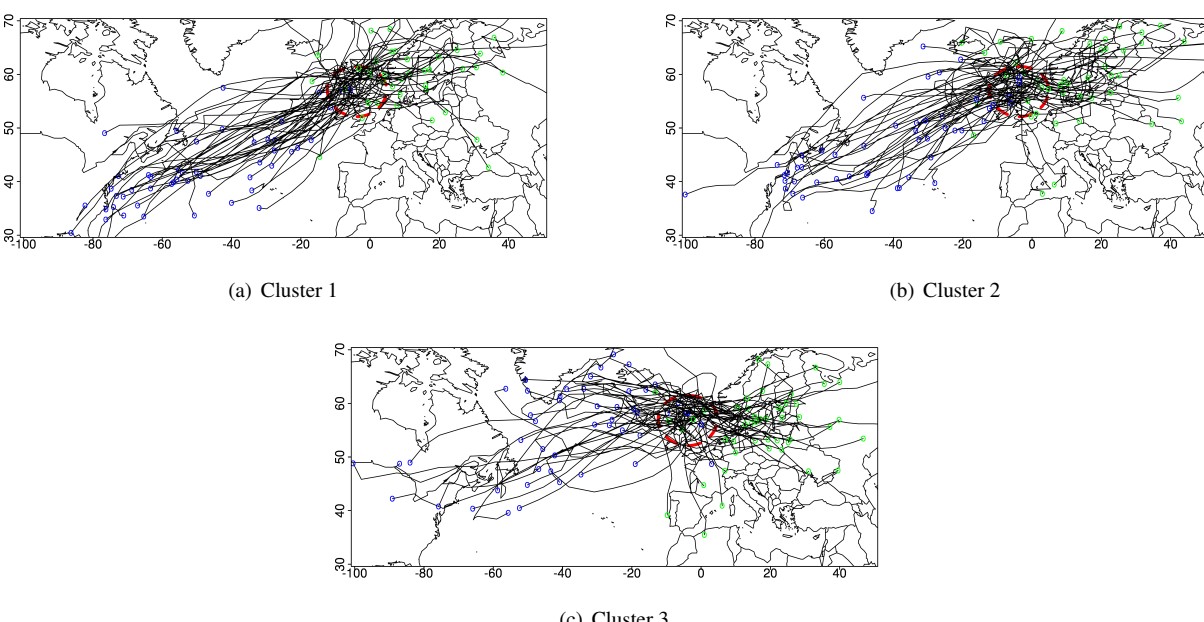

(a) Cluster 1

(b) Cluster 2

(c) Cluster 3

**Figure 8.** The three clusters that were identified by the clustering probabilistic technique for intense cyclones affecting the British Isles. Cluster 1 (top left) with its southwest to northeast progression, Cluster 2 (top right, west to east) and Cluster 3 (bottom, northwest to southeast). 50 random tracks are shown for each of the clusters. The blue points mark the beginning of the track whereas the green ones depict the last time step of each trajectory. The red circle encloses the British Isles and has to be crossed at least once in the life cycle of the ETC.



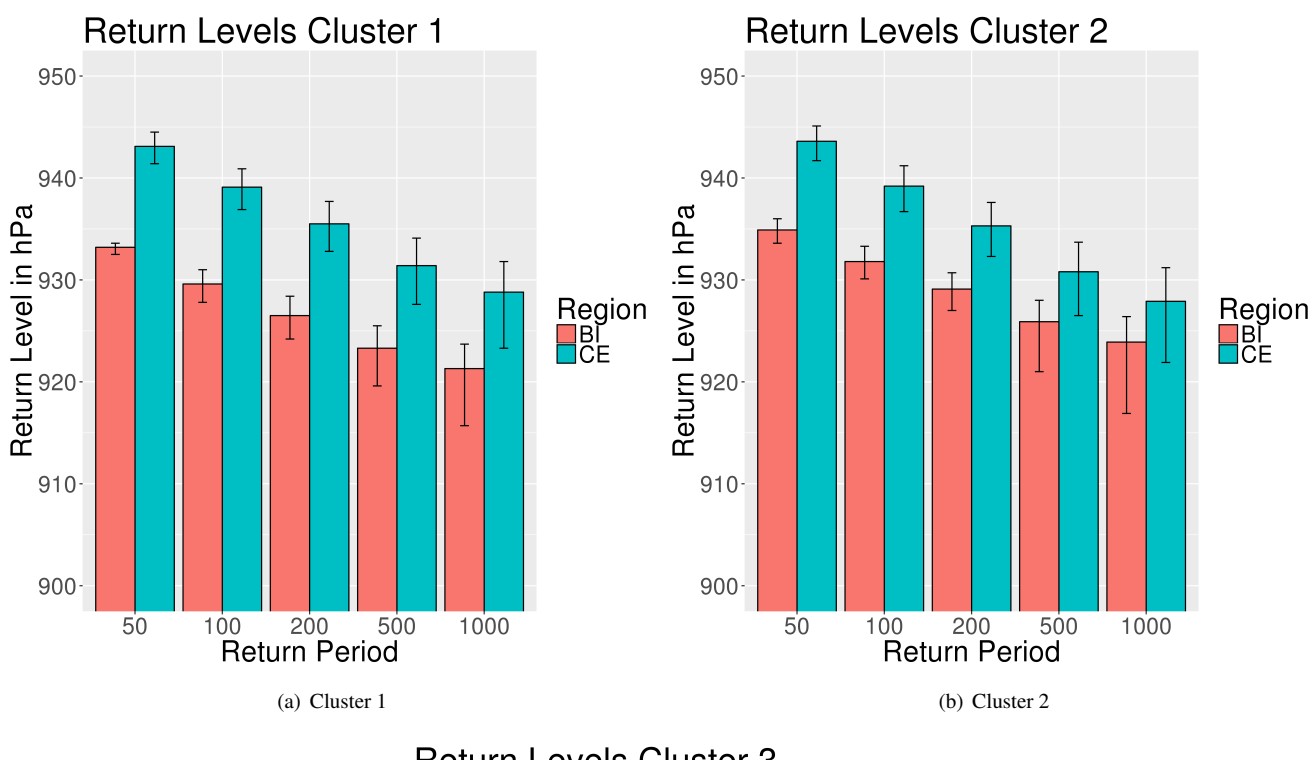

(a) Cluster 1    (b) Cluster 2

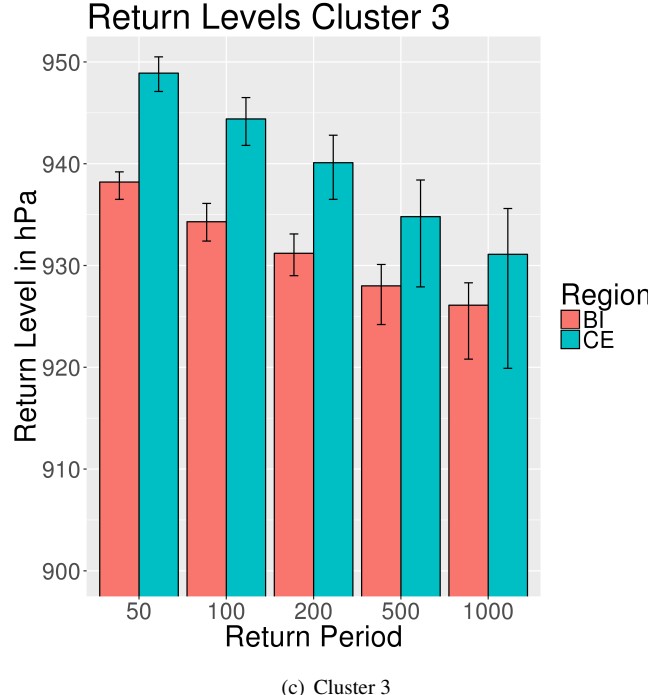

(c) Cluster 3

**Figure 9.** Weighted return levels of minimum core pressure for given return periods of the British Isles and Central Europe. The 95% confidence intervals are calculated via the profile likelihood method and are presented in brackets. The arrow behind the clusters represents the general direction of the cyclone progression as discussed. The last row provides an estimate of a lower bound of minimal core pressure.


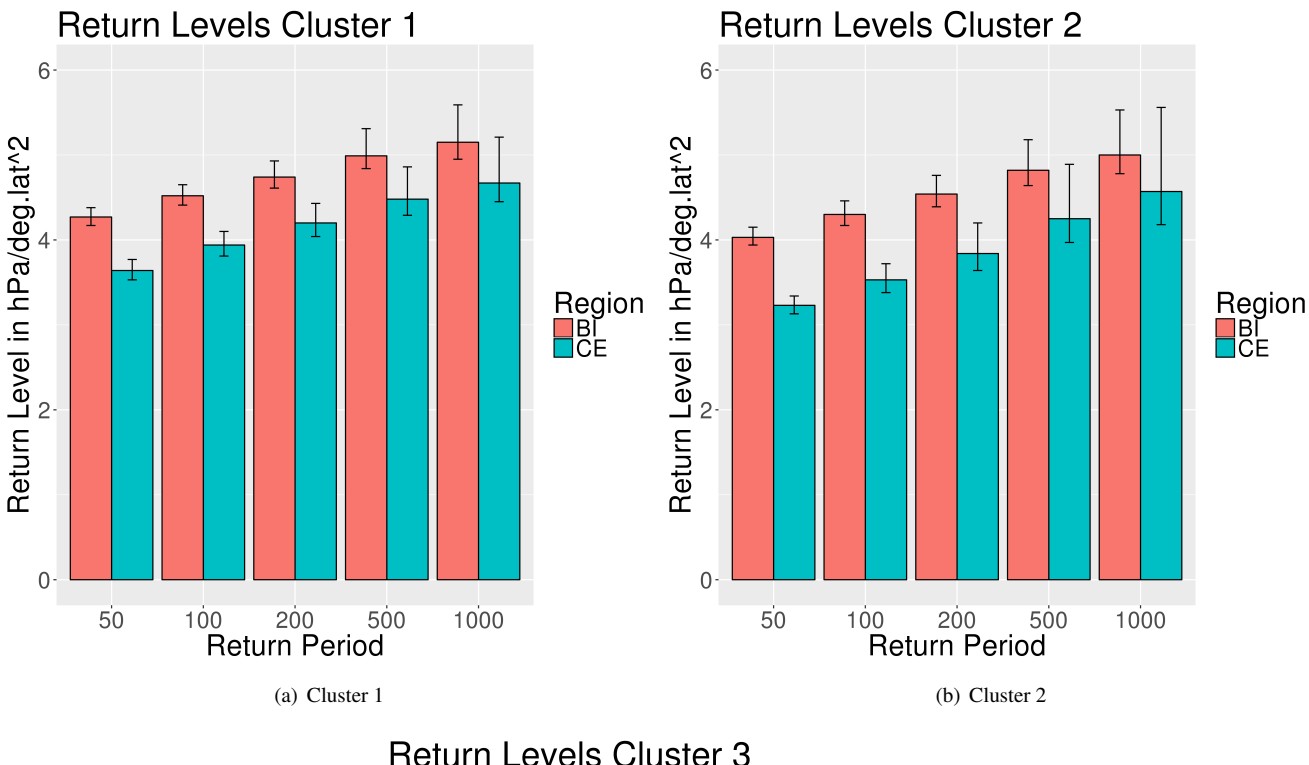

(a) Cluster 1

(b) Cluster 2

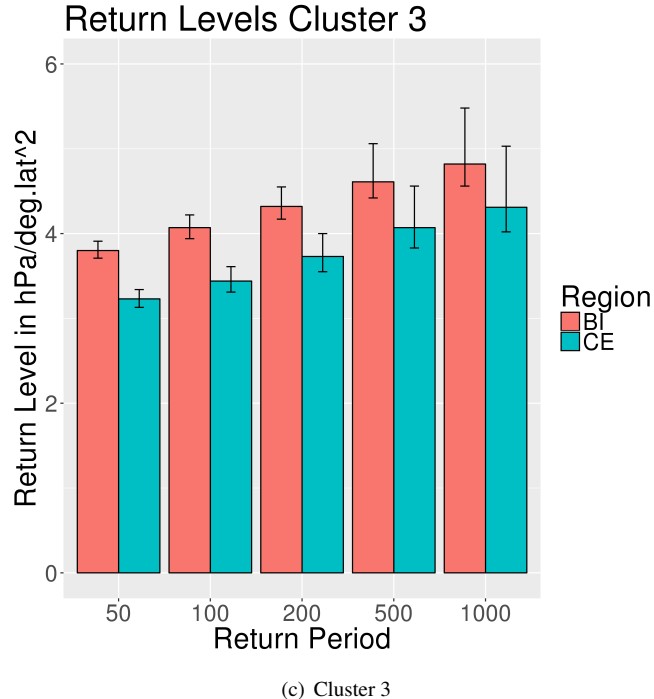

(c) Cluster 3

**Figure 10.** Weighted return levels of curvature for given return periods of the British Isles and Central Europe. The 95% confidence intervals are calculated via the profile likelihood method and are presented in brackets. The arrow behind the clusters represents the general direction of the cyclone progression as discussed. The last row provides an estimate of a lower bound of minimal core pressure.