# Peer review of "Spatial variability and potential maximum intensity of winter storms over Europe"

_Natural Hazards and Earth System Sciences, 2018_

## Referee Comment (RC1) · Anonymous Referee #1 · 8 Jan 2019

In this paper an attempt is made to identify windstorms affecting certain regions of central and northern Europe with the aid of ECMWF operational seasonal forecast system 4, consisting of 51 members of retrospective forecasts each resemble an artificial reality until November 2017. The paper presents some scientific interest, considering previous related work. More specifically I have some major scientific queries:

1. I cannot understand the advantage of employing ECMWF operational climatic predictions (system 4) to study the windstorms on a climatological basis and not reanalysis datasets, such as the ERA-20C at a similar resolution? Recognizing the merit of the great data amount, how the authors are confident about the reliability of these

datasets? 2. Following the same comment, I cannot understand statement in page 3 "That way the ensemble serves as a unique data archive which can be used to assess the statistical uncertainty more precisely compared to exploiting observational reanalysis data for this kind of estimation". The authors should clarify and verify this point, since determines the novelty of the paper as compared to previous related studies. In any case, this statement is not discussed or verified in section 4. 3. Similarly, the authors should clarify the statement in page 2: "Clearly none of the windstorms found in these forecasts ever happened, however each of them represents one possible physical consistent realisation of a potential reality" 4. In section 2: "31 years of data, is equivalent to 1581 "virtual" years". Please clarify. 5. The spectral resolution of the data is T255. Is this the same with reanalysis data ERA-interim or ERA-20C? 6. In section 2: "...This is implemented by only taking into account windstorms that affect a country at least once in their lifetime, i.e. by defining a radius around a country/an area through which a windstorm or a cyclone has to pass". Please clarify. How this radius is defined? Is it defined a priori? 7. Section 2: As the cyclones identified by the Murray and Simmonds algorithm are not necessarily extreme in terms of impact, the minimal core pressure and the maximum curvature of an identified track both have to be within the lowest respectively highest 5% of all tracks at least once within the defined radius" I think this percentile is somewhat arbitrary. Is it based on statistical analysis? Or in previous studies? 8. What do you mean by "local maximum" or local 98th percentile" in practice? At every grid point? 9. In section 3: apart from central pressure and curvature as measures of the cyclone intensity, the local pressure drop is an important measure that determines intense and mainly explosive cyclones that are responsible for wind storms. 10. Section 4: From Table 1, I assume that the 3 clusters are identical for all 4 regions. Is this true? For this reason the clusters are displayed for the two regions ?

Other comments 1. Abstract: Main findings are missing. A large part is devoted to explain the advantages of using seasonal forecasts. This is not the scope of the abstract. 2. Section 1: why the ETCs responsible for windstorms in Europe have their

origin over the NW Europe? A reference is required. 3. Section 1: "..they usually follow an eastward trajectory". Similar comment 4. Section 1 is not structured in paragraphs. 5. Section 2 is not structured in paragraphs. For instance, in page 3, line 13 a new paragraph could start with the statement "Two different types..." 6. Since section 2 is entitled "Data" , the authors should focus only on the data used. The remaining part refers to methodology that is described in section 3. Therefore, section 2 and 3 should be reformulated 7. Section 2, page 4, line 4: "Due to the constraint for the tracked cyclones.." What do the authors mean? 8. Section 5 is not structured in paragraphs. For instance, in page 12 at lines 14, 24 and page 13 at lines 7, 23 9. Legend of Figure 1: Replace "Km2" by upper case "Km2"

---

## Referee Comment (RC2) · Anonymous Referee #2 · 9 Jan 2019

The study uses operational ECMWF seasonal ensemble forecast data to investigate the most intense extratropical cyclones by means of extreme value statistics. The impact of the cyclones is evaluated by using the windstorm index SSI and different regions within Europe are considered. The paper addresses a relevant topic that meets the interests of NHESS. Using this large data set of a state-of-the-art NWP model is an innovative approach and the method applied is appropriate.

From my view the manuscript needs a major revision in terms of carefulness: (1) The study uses regions (SC, CE) that are not shown nor described in terms of position and size (section 3, p.4). Additionally, the choice of the regions is not motivated. (2) In

section 4.2, p.9 one figure is referenced that is not in the paper. (3) Sections 1, 2, 4.1, 4.2 and 5 must be divided into paragraphs to make them readable.

Other comments: (4) Sentence 2 of the abstract should be deleted because it does not express what you have done in the study. (5) p.1, line 15, 16: From my understanding, "somewhere" and "about" sound a bit too sloppy to describe amounts of losses and casualities. There should be a more exact reference. (6) p. 3, line 8: I find "years" confusing in this context. (7) p. 3, line 9, 10: What do you mean by "observational" reanalysis data? (8) section 2: What times/time resolution of the data set have you used? (9) Fig. 1: What is the unit and the depicted level of the wind speed? (10) Fig. 1: Here, you could show the circles for all regions. (11) p.4, line 1: As above, replace "years" by seasons. (12) p. 6, line 21: In the beginning of the main section: What are the "other two regions" here? (13) Fig. 2,3: The red circle is hardly visible, can you draw it on top of the trajectories? (14) p. 9, lines 5ff: The heights of the bar figures 4,9 and 10 are barely comparable among the panels. A number on top of each bar could help, or a finer resolution of the horizontal lines. In addition, there is the unit missing for the return period. (15) p.9: Same as above: The long text needs some paragraphs. (16) p.9, lines 26ff: How are celerity and duration defined, how are they used to construct Fig. 5? (17) p.9, lines 34ff: This can not be understood without the figure. (18) p.9, line 32-34: This statement is related to which region? (19) p. 10, line 1ff, Fig. 6,7: How exactly are the composites constructed, which times of the windstorm are used? (20) Same, more scientific: What do you want to address with the composites? Your argumentation goes in two different directions: Do they represent the cyclone related to the windstorm or the steering flow responsible for the trajectories? For the steering flow the 700hPa geopotential is a more appropriate field. Still, the composites show a climatological picture. Are the windstorms of smaller scale embedded somewhere in the Islandic low? Then, it would be interesting to show them as a disturbance field or high frequency field where the climatological low pressure system is subtracted. If there are multiple time steps for each cyclone/windstorm you could consider to use only one each, e.g. the most intense or the one when entering the region. Please revise this

paragraph. (21) p. 10, line 26: Delete the "a". (22) p. 10, lines 29 till p.11, line 5: This section is too long. What about drawing a marker at the beginning of the windstorm identification above each trajectory? (23) p. 11, table 2: Is there a physical reason why the Germany/Benelux region should be affected by a potentially deeper cyclone than the British Isles (which are closer to the Islandic low)? This is also in contrast to the results in Fig. 9. (24) section 4.2 and Fig. 10: Wouldn't it be more meaningful to show the statistics of the wind speed instead of cyclone curvature? (25) p. 12, line 11: Again, you do not have 1500 "years" of data. (26) p.13, lines 3-5: How is that meant? (27) p.13, line 3 16,17: What do you mean: "cyclones ... are ... lower"?

---

## Author Comment (AC1) · 15 May 2019

We would like to thank the reviewer for their valuable comments on our study. In the following we would like to address every comment individually and present our opinion and also indicate changes that have been made.

1. I cannot understand the advantage of employing ECMWF operational climatic predictions (system 4) to study the windstorms on a climatological basis and not reanalysis datasets, such as the ERA-20C at a similar resolution? Recognizing the meritof the great data amount, how the authors are confident about the reliability of these datasets?

[Figure]

The great merit of using the seasonal hindcast ensembles is the shear amount of data. Reanalysis products (like ERA-20C) only entail around 100 winter seasons, whereas with the seasonal hindcasts we obtain more than 1500 seasons. There have been studies investigating the quality of wind(storms) in ECMWF System 4 (e.g. Befort et al., 2018 or Walz et al., 2018b). The overall conclusion is that despite the reliability not being perfect they feature skill in year-to-year variability and spatial coherence. The reliability of reanalysis products, especially in the beginning of the observation period, is also not entirely given. Befort et al. (2016) for example found strong trends for the first 50 years of windstorms in ERA-20C. We have added a bit of explanation why we think the seasonal hindcasts represent a great addition to climatological data and hope this answers the reviewers question

2. Following the same comment, I cannot understand statement in page 3"That way the ensemble serves as a unique data archive which can be used to assess the statistical uncertainty more precisely compared to exploiting observational reanal-ysis data for this kind of estimation". The authors should clarify and verify this point,since deter-mines the novelty of the paper as compared to previous related studies. In any case, this statement is not discussed or verified in section 4.

Similar to above, just based on the pure amount of data statistical inferences can be made more easily. We have added more explanation in the text, especially in section 4.

3. Similarly, the authors should clarify the statement in page 2: "Clearly none of the windstorms found in these forecasts ever happened, however each of them represents one possible physical consistent realisation of a potential reality"

We are not entirely sure what the reviewer wants to be clarified here. All the storms that we could track in the data are "fictional". Some might be similar to a storm that has already happened in reality, some might happen in a similar way in the future. The seasonal forecast itself however does not try to "replicate" storms that happened in

reality.

4. In section 2: "31 years of data, is equivalent to 1581 "virtual" years". Please clarify. Changed years to winter seasons. 31 years times 51 members equals 1581 trackable winter seasons.

5. The spectral resolution of th edata is T255. Is this the same with reanalysis data ERA-interim or ERA-20C?

Same as ERA-interim.

6. In section 2: "...This is implemented by only taking into account windstorms that affect a country at least once in their lifetime, i.e. by defining a radius around a country/an area through which a windstorm or a cyclone has to pass". Please clarify. How this radius is defined? Is it defined a priori?

The radius is defined a priori by choosing a centre point within a country and setting the radius so that it encompasses the entire country. Naturally small bits of other countries are included sometimes, this, however, does not pose a problem as a storms also do not care about political borders. We have added a figure in the supplements showing the radii for the different regions.

7. Section 2: As the cyclones identified by the Murray and Simmonds algorithm are not necessarily extreme in terms of impact,the minimal core pressure and the maximum curvature of an identified track both have to be within the lowest respectively highest 5% of all tracks at least once within the defined radius" I think this percentile is somewhat arbitrary. Is it based on statistical analysis? Or in previous studies?

In a way it is arbitrary. However it represents a good compromise between choosing enough events to still be able to produce significant statistical analysis and to choose MSLPs that only occur only in every 20th on average. The 95th (or 5th) percentile has been used in previous studies as an indicator of "extremeness" (e.g. Walz et al., 2018b or Della-Marta et al., 2009)

8. What do you mean by "local maximum" or local 98th percentile" in practice? At every grid point? Yes it is a grid point based approach. So the tracking is based on the local 98th percentile to account for geographically different wind climatologies. Please refer to Leckebusch et al. (2008) for more details.

9. In section 3: apart from central pressure and curvature as measures of the cyclone intensity, the local pressure drop is an important measure that determines intense and mainly explosive cyclones that are responsible for wind storms.

Indeed that would also be an interesting thing to investigate as it has been done in numerous studies before. However we decided to focus on the two characteristics. We have changed (or transformed) the curvature however into means of a Rossby number to make the numbers a bit more tangible as especially the unit of the curvature seems very abstract.

10. Section 4: From Table 1, I assume that the 3 clusters are identical for all 4 regions. Is this true? For this reason the clusters are displayed fort he two regions?

You are correct, the three found clusters are identical for all the regions, that is why we chose only two show results for the two presented areas. We have added that in the manuscript.

Other comments 1. Abstract: Main findings are missing. A large part is devoted to explain the advantages of using seasonal forecasts. This is not the scope of the abstract. Changed the abstract.

2. Section 1: why the ETCs responsible for windstorms in Europe have their origin over the NW Europe? A reference is required. Added a reference.

3. Section 1: "..they usually follow an eastward trajectory". Similar comment See above.

4. Section 1 is not structured in paragraphs. The overall structure was not great, we have added paragraphs throughout the manuscript.

5. Section 2 is not structured in paragraphs. For instance, in page 3, line 13 a new-paragraph could start with the statement "Two different types..." The overall structure was not great, we have added paragraphs throughout the manuscript.

6. Since section 2 is entitled "Data" , the authors should focus only on the data used. The remaining part refers to methodology that is described in section 3. Therefore, section 2 and 3 should be reformulated

7. Section 2, page 4, line 4: "Due to the constraint for the tracked cyclones.." What do the authors mean? This is a reference to the 95% (5%) constraint of only using the top 5% of extreme MSLP and curvature/Rossby number. We have changed the wording accordingly.

8. Section 5 is not structured in paragraphs. For instance, in page 12 at lines 14, 24 and page 13 at lines 7, 23

Changed the formatting.

9. Legend of Figure1: Replace "Km2" by upper case "Km2"

Changed it.

---

## Author Comment (AC2) · 15 May 2019

We would like to thank the reviewer for their valuable comments on our study. In the following we would like to address every comment individually and present our opinion and also indicate changes that have been made.

(1) The study uses regions (SC, CE) that are not shown nor described in terms of position and size (section 3, p.4). Additionally, the choice of the regions is not motivated.

We have added a figure that shows the circles around all the regions as a supplement. Furthermore we have added some description on how the radii were defined. Basically

it is just choosing the "centre" of a region and selecting a radius to encompass the respective country/area.

(2) In section 4.2, p.9 one figure is referenced that is not in the paper. The reference was amended accordingly.

(3) Sections 1, 2, 4.1,4.2 and 5 must be divided into paragraphs to make them readable. There was some Latex formatting error, apologies for that. Paragraphs have been added to the entire manuscript.

Other comments: (4) Sentence 2 of the abstract should be deleted because it does not express what you have done in the study. The sentence was deleted, the entire abstract was revised also.

(5) p.1, line 15, 16: From my understanding,"somewhere" and "about" sound a bit too sloppy to describe amounts of losses and casualities. There should be a more exact reference. Fully agreed, deleted the two words. Overall we are trying to be more precise.

(6) p. 3, line 8: I find "years"confusing in this context. We have changed it to winter seasons. Of course it is not years per se

(7) p. 3, line 9, 10: What do you mean by "observational"reanalysis data? This was to stress that we consider reanalysis as observations, despite them being a model as well. Removed the observational in order to avoid confusion.

(8) section 2: What times/time resolution of the data set have you used? Added the description on time resolution (6-hourly data).

(9) Fig. 1: What is the unit and the depicted level of the wind speed? Added the description in the legend. We used 10-m wind speeds in m/s.

(10) Fig.1: Here, you could show the circles for all regions. We think that the plot would get too busy here. We have added a figure in the supplementary figures to show the

circles.

(11) p.4, line 1: As above, replace"years" by seasons. Done.

(12) p. 6, line 21: In the beginning of the main section: What are the "other two regions" here? The other two regions are Scandinavia and Central Europe. The latter is quite similar to Germany and the Benelux, it includes France additionally. Please refer to the figure in the supplement.

(13) Fig. 2,3: The red circle is hardly visible, can you draw it on top of the trajectories? As we have added an additional figure we hope that this suffices as reference to where the circles are located.

(14) p. 9, lines 5ff: The heights of the bar figures 4,9 and10 are barely comparable among the panels. A number on top of each bar could help,or a finer resolution of the horizontal lines. In addition, there is the unit missing for the return period.

(15) p.9: Same as above: The long text needs some paragraphs. Done.

(16)p.9, lines 26ff: How are celerity and duration defined, how are they used to construct Fig. 5? Some description was added on page 6 lines 30ff. Celerity as the translational speed between two 6-hour time steps and the duration in days, thus number of 4 time step blocks in the tracked windstorm.

(17) p.9, lines 34ff: This can not be understood without the figure. The comparison with the other region was removed. This part was remaining from a previous version of the manuscript. Apologies for that.

(18) p.9,line 32-34: This statement is related to which region? See above. Removed that statement.

(19) p. 10, line 1ff, Fig. 6,7:How exactly are the composites constructed, which times of the windstorm are used? There is some description around lines 33ff on page 6. All time steps of windstorms for a respective cluster are averaged to create the

composites.

(20) Same, more scientific: What do you want to address with the composites? Your argumentation goes in two different directions: Do they represent the cyclone related to the windstorm or the steering flow responsible for the trajectories? For the steeringflow the 700hPa geopotential is a more appropriate field. Still, the composites show a climatological picture. Are the windstorms of smaller scale embedded somewherein the Islandic low? Then, it would be interesting to show them as a disturbance field or high frequency field where the climatological low pressure system is subtracted. If there are multiple time steps for each cyclone/windstorm you could consider to use only one each, e.g. the most intense or the one when entering the region. Please revise this paragraph.

We appreciate the comment and see the valid point here. Due to data availability and facilities we are unfortunately not able to do further analysis on other data fields. We rephrased the paragraph so that the MSLP composites can be seen as a "proxy" the steering flow. As they indeed show the correct flow conditions when assuming a geostrophic wind. We hope that this answer is satisfactory for the reviewer.

(21) p. 10, line 26: Delete the "a". Done.

(22) p. 10, lines 29 till p.11, line 5: This section is too long. What about drawing a marker at the beginning of the windstorm identification above each trajectory? We shortened the paragraph. We believe the markers would make the "spaghetti plots" even more busy so we decided not to add markers.

(23) p. 11, table 2: Is there a physical reason why the Germany/Benelux region should be affected by a potentially deeper cyclone than the British Isles That is a valid point however one has to keep in mind that these estimates are purely based on statistical assumptions within the extreme value theory, thus they are probably not "physical" meaningful any way. If you look at the return period plots it is obvious that cyclones affecting the BI are considerably deeper than the ones for CEBE. Added this to the

manuscript as well.

((24) section 4.2 and Fig. 10: Wouldn't it be more meaningful to show the statistics of the wind speed instead of cyclone curvature? Decided to transform the curvature to a version of the Rossby number to show the extremeness of the cyclones with regards to the curvature.

(25) p. 12, line 11:Again, you do not have 1500 "years" of data. Change it. See above

(26) p.13, lines 3-5: How is that meant? This sentence was a remainder from an earlier version of the manuscript and was thus removed.

(27) p.13, line 3 16,17: What do you mean: "cyclones ... are ... lower"? Should have read deeper. Changed it.
* * *
[Figure]

**Fig. 1.** Circles to define the areas through which a windstorm has to pass for the respective regions. Black for the BI, green for GEBE, blue for SCAN and red for CE.

[Figure]

**Fig. 2.** Circles to define the areas through which a cyclone has to pass for the respective regions. Black for the BI, green for GEBE, blue for SCAN and red for CE.

---

## Author Comment (AC3) · 15 May 2019

Michael A. Walz and Gregor C. Leckebusch

michael_walz@swissre.com

Figures that define the selection radii for the different regions.

[Figure]

[Figure]

**Fig. 1.** Circles to define the areas through which a windstorm has to pass for the respective regions. Black for the BI, green for GEBE, blue for SCAN and red for CE.

[Figure]

**Fig. 2.** Circles to define the areas through which a windstorm has to pass for the respective regions. Black for the BI, green for GEBE, blue for SCAN and red for CE.